# Perceived Quality and Users’ Satisfaction with Public–Private Partnerships in Health Sector

**DOI:** 10.3390/ijerph19138188

**Published:** 2022-07-04

**Authors:** João M. S. Carvalho, Nuno Rodrigues

**Affiliations:** 1Department of Economics and Management, REMIT—Universidade Portucalense, R. António Bernardino de Almeida 541, 4200-072 Porto, Portugal; 2Social Sciences Institute, CICS.NOVA—Universidade do Minho, 4710-057 Braga, Portugal; 3Department of Social Sciences and Management, CEG—Universidade Aberta, 1250-100 Lisboa, Portugal; 4Instituto Politécnico de Gestão e Tecnologia, 4400-107 Vila Nova de Gaia, Portugal; nuno.rodrigues@islagaia.pt

**Keywords:** public–private partnerships, public management hospitals, quality of health services, users’ satisfaction, literacy level

## Abstract

In Portugal, the government has accepted private management within public hospitals since 1996. The objectives of the state were to ensure more efficiency in resource management and maintain or increase the service quality provided to the users. Four public hospitals have been managed with a public–private partnership (PPP) approach. This study aimed to empirically analyse the degree of satisfaction of the Portuguese population regarding the service quality provided by PPP and Public Management Hospitals (PMH) within a structural equation model, and verify if people’s literacy level, age, education, and income moderate their opinions. The study used 2077 valid questionnaire responses applied in the four regions served by the eight hospitals. The results show that the users of the PPP hospitals are more satisfied than those from PMH with statistical significance. Literacy level moderates the relationship between perceived quality and users’ satisfaction, and education moderates the same relationship only in the context of PPP hospitals. More educated people with a high literacy level are more demanding, both regarding PPP and PMH hospitals. Nevertheless, the results are very beneficial to the PPP model; thus, improved decision-making regarding contract renewal might help policymakers consider the findings of this paper.

## 1. Introduction

Public–Private Partnerships (PPP) consist of an agreement between two structurally different organisations with different strategies and operational objectives [1,2]. While partners belonging to the private sector aim to maximise profit margins or improve business performance, partners in the public sector aim to optimise social, political, and budgetary objectives [3]. However, PPP relationships differ fundamentally from the conventional public model. In a conventional public contracting model, the parts predict and assume risks. In the PPP model, the development of risk-sharing mechanisms is the key to increasing returns for both the public and private sectors [4].

PPP theory focuses on a clear definition of outcomes, operating around a detailed business case and a thorough project development phase, putting project and contract management plans in place, and involving market testing at various levels [5]. The predominant characteristic of PPP models focuses on the public sector contracting or purchasing a set of services or goods rather than building/buying or supplying them [2].

OECD [6] (p. 17) defined PPP as an agreement between the public sector and one or more private entities. The latter provides a service that meets the requirements defined by the government and, at the same time, generates profit for shareholders; these two requirements depend on the risks allocated to each party. The literature defines several typologies applicable to the PPP model. Nevertheless, based on contract, the Design–Build–Finance–Operate (DBFO) model has been a general approach [7]. In this model, the public partner defines the services it wants the private sector to provide, leaving the private sector to finance and build the committed asset. In addition to construction, the private partner manages the asset and provides its services [8].

The PPP model has become increasingly popular in recent years, including in the health sector. This situation is due to the strategic change in private health organisations, which have recognised the importance of public health objectives to their medium and long-term goals and accept a broader view of social responsibility as part of business strategy [9,10]. The support provided by the World Health Organization, by playing a significant role in health policy formulation and health care standard-setting and by welcoming and encouraging state–private partnerships in health care financing, service delivery, and research, has also contributed to this growth in popularity [11].

In many countries, hospitals generate about one-third of health expenditure, so even a tiny growth rate in the hospital sector significantly affects total health expenditure [12]. The costs allocated to hospital infrastructure are of great importance because some countries have a delay in developing their health infrastructure (infrastructure gap) [12], meaning that hospitals in these countries will need to renew their infrastructure over the next few years. Therefore, if governments wish to reduce spending in the health sector, it is crucial to analyse and optimise hospital spending and their management models [13]. Regarding health service delivery, advances in information technology have increased patients’ knowledge and expectations about the quality of health care, forcing service providers to seek new ways of delivering this care to meet users’ expectations [14]. The effective development of public health is only possible through constant investments and innovations, driving the use of the PPP model. For instance, in Russia, it led to a reduction in mortality, an improvement in the quality of medical care, and an increase in life expectancy [15].

In addition to the open debate and controversy present in the scientific community e.g., [16,17,18], given the most recent studies, there is a significant gap in knowledge due to the shortage of research analysing the impact of the PPP model concerning service quality and satisfaction. As Hellowell [17] argued, there is little guidance for decision-makers regarding the circumstances in which the PPP model is expected to produce good results. In this sense, Petersen [19] appeals for a greater academic focus on rigorous evaluations of the PPP model, with prominence on quality as a performance measure. This idea follows the conclusion of the Portuguese Audit Office Court [20], which states that some of the future challenges essentially consist of monitoring and adapting performance indicators in all National Health Service (NHS) hospitals based on the experience gained. This report [20] presented other aspects, such as costs and value for money (a measure of the utility of the money spent), which were favourable to PPP hospitals. However, they defended those other mechanisms such as the evaluation of user satisfaction should also be implemented. Already in 2019, the Audit Office Court (AOC) mentioned that ‘the State did not consider the user satisfaction aspect in the overall assessment given the inexistence of comparable information obtained by the Ministry of Health in publicly managed hospitals of the SNS’ [21] (p. 19). Additionally, the AOC recommended that the state conduct and publicise user satisfaction surveys about all NHS hospital units, comparing the results among PMH and PPP hospitals [21].

Healthcare is one of the fastest-growing sectors in the service economy [22]. User satisfaction and service quality are critical factors in the strategic planning of healthcare units’ management processes. The extent of public and private sector responsibilities may vary considerably between the various typologies of the PPP model. However, the public sector retains responsibility for deciding on the services to be provided, the quality and performance standards of those services, and taking necessary corrective action if quality or performance falls short of expectations [23]. Health service quality is traditionally based on professional practice standards. However, patient perceptions of health care have been predominantly mentioned as a crucial indicator for measuring health care quality and as a critical component of improving clinical performance and effectiveness [24,25].

Donabedian [25] argued that quality in health is a judgement about the technical component and the positive characteristics of interpersonal relationships between the patient and the professionals, distinguishing it into three major components: (a) technical aspects—referring to the medical knowledge in diagnosis and treatment; (b) interpersonal aspects—the ability to interact with patients; and (c) affability aspects—including the comfort conditions of the health units. The same author stated that quality could be divided into two aspects: the technical quality and the quality perceived by patients. Technical quality concerns the provided services quality, i.e., it is related to the competence of the diagnosis made by health professionals and the treatment outcome. On the other hand, the quality of the patients’ perception is related to their satisfaction with the health care provided [26]. Thus, using clinical outcomes of healthcare provision as a criterion for quality assessment brings advantages, and its validity as a dimension of quality is rarely questioned [27].

Due to the private sector’s involvement, government assets and intellectual property can be used more productively, leading to a substantial improvement in public facilities and services [28]. Additionally, through the appropriate use of the private sector’s skills, experience, technology, and innovation, public services can be more satisfactorily delivered [1,29], which may translate into intentional patient behaviour to return and recommend a given healthcare facility [30]. Users’ reluctance may affect the perception of service quality [31], which in turn critically influences the choice of healthcare providers [32] and users’ choice of hospitals [33]. This idea is supported by Skietrys et al. [34]. They argued that participants in PPPs are required to accept new values, such as controlled competitiveness, an emphasis on the population’s needs, and the implementation of accountability and quality systems. Thus, community well-being should be at the forefront of modern management [35].

In the case of PPP hospitals, user satisfaction becomes even more critical considering fulfilling one of the requirements of PPP projects—obtaining social benefits for the population. Now, the theory tells us that there is a direct relationship between the quality of service provided and user satisfaction, which can be seen from two perspectives: the first one states that a satisfied user leads to a good perception of service quality [36], and the second recommends that service quality leads to user satisfaction [37,38]. These perspectives confirm the strong correlation between user satisfaction and service quality [39,40]. In addition, other authors have linked service quality to behavioural intention [41,42].

The literature defines service quality as a global judgment or attitude related to the overall excellence or superiority of the service [43]. Additionally, service quality is defined as assessing a user’s overall service quality by applying a disconfirmation model, i.e., the gap between service expectations and actual performance [37,44]. The literature also identifies three reasons for measuring user satisfaction: (1) user satisfaction is the primary goal of the health care provider; (2) user satisfaction provides valuable data about the structure, process, and outcomes of health care; and (3) satisfied and dissatisfied users have various behavioural intentions [45]. There is empirical evidence supporting a causal link between health service quality and user satisfaction, e.g., [46,47]. The user’s perception of the basic medical service positively influences service quality and users’ trust and satisfaction [48].

Within the scope of the present research, it becomes clear that measuring user satisfaction consists of the main objective of PPP hospitals which have to provide an equivalent or superior quality of service to PMH. Moreover, users’ satisfaction significantly affects the reputation of hospitals in the community [49]. In addition to evaluating user satisfaction, it is also essential to assess the moderating effect of the population’s health literacy on their perception of health service quality in PPP hospitals. The degree of literacy plays a crucial role when we know that one of the success factors of PPPs is social acceptance, i.e., the population’s acceptance of the delivery of public service to a private partner [50]. In this context, it becomes even more imperative to conduct research that compares the literacy level with empirical results regarding the quality of service provided and the degree of user satisfaction.

Nutbeam [51] defines health literacy as the set of cognitive and social skills that determine the motivation and ability of individuals to obtain, understand, evaluate and apply information to maintain and promote a healthy life. The International Adult Literacy Survey defines literacy proficiency levels in using the information to function in society and the economy. According to the National Library of Medicine, health literacy is defined as how individuals can obtain, process, and understand health information and services and make the necessary decisions. According to Doyle et al. [52], there are five significant advantages in the face of a high degree of health literacy: (1) self-efficacy; (2) health promotion; (3) assertive use of health services; (4) disease self-management; and (5) empowerment. However, while users have the right and are effectively encouraged to make active choices, for example, by reviewing comparative quality information on healthcare provider performance, few do so. Fotaki et al. [53] conclude that the choice between hospitals is not a priority for the general public in the UK and the US. Many patients (34–70%) rely on their family physician’s choice of referral to a specific hospital. For relatively simple procedures, they prefer to go to the nearest hospital [54,55]. Literacy tends to play a crucial role when we know that one of the success factors of PPPs is social acceptance, i.e., the acceptance by the population of the delivery of public service to a private partner [50].

Consequently, it seems very relevant to address the issue of PPPs within the scope of health in Portugal. Moreover, the gaps identified in the literature are substantial. For example, ref. [19] indicated that most studies in the literature do not examine service quality. Osei-Kyei and Chan [56] pointed to the need to assess the potential limitations in applying the PPP model. Furthermore, Roehrich et al. [57] argued that the current empirical evidence on the benefits of PPPs presents mixed results. The current literature also indicates that despite the increasing attention paid to the effectiveness of PPPs in the health sector, few studies provide a practical framework that enables the evaluation of PPP effectiveness, mainly because there is no empirical evidence about the actual effectiveness of PPP projects [10].

Thus, our objectives are to empirically verify the satisfaction of the Portuguese population regarding the PPP model implemented in four Portuguese hospitals and similar PMH; to assess the quality of service provided by hospitals operating on the PPP model in Portugal, and to analyse the possible moderator effect of several variables (literacy level, age, education, and income) of the Portuguese population regarding the PPP model in health. Thus, this study aims to answer the following questions:How satisfied is the Portuguese population with the quality of service provided by PPP hospitals?Is there any difference in the users’ satisfaction between PPP hospitals and PMH?Is the proposed model adjusted to data collected among the Portuguese population?Are literacy level, age, education, and income moderator variables of the relationship between quality perceived by the users and their satisfaction?

These research questions are even more important because the negative results of a given PPP project led to dissatisfaction among the population and a negative perspective of social well-being [58]. To answer these specific questions, we designed and applied questionnaires to a sample of the Portuguese population served by the existing four PPP hospitals and PMH hospitals with the same characteristics in the same regions.

## 2. Methods

This study used a quantitative approach through a survey among the population served by the target hospitals. The scales that measure the quality of hospitals’ services (Healthqual) and the level of literacy of the Portuguese population about the PPP management model were assessed and validated. Based on the known relationships among the variables, a model was proposed to be analysed through structural equation modelling.

There were four cases of PPP hospitals, namely in neighbouring municipalities of Lisbon: Cascais (5th large municipality), Loures (6th), and Vila Franca de Xira (17th), and in the North of the country, Braga, that is the seventh large municipality of Portugal. They also studied patients’ opinions from four other hospitals in the same regions with a public management model.

### 2.1. Sampling

Comparing PMH and PPP hospitals is a complex exercise, mainly due to their heterogeneity. However, there are defined hierarchical clusters by the Central Administration of Health Systems (CAHS) according to their characteristics, namely the type of hospital (central, regional, etc.), the type of services provided (medical specialties), population coverage, and principal component analysis of their standardised cost variables [59]. This hierarchical clustering model does not have a group A. The others were categorised as follows: B, C, D, E, and F. Only groups B, C, and D were relevant for this study, as they contain the four PPP hospitals: Vila Franca de Xira (B), Cascais and Loures (C), and Braga (D). Thus, and considering the need to compare distinct management models, each hospital will be compared with other hospitals with similar characteristics, meaning hospitals belonging to the same group.

The chosen sample includes the PMH of Figueira da Foz (Group B), Leiria and Setúbal (C), and Évora (D), which are the most similar to the PPP hospitals considering the population coverage of each hospital and the unit type. Thus, the population of PMH under study refers to all hospitals included in the hierarchical groups B, C, and D because they correspond to the clusters where exist PPP hospitals with the same characteristics. Additionally, the choice of the 4 PMH hospitals took into account the official study conducted by the Portuguese Healthcare Entity [60], where a set of 33 hospitals (including PPP and PMH) was defined, including the 4 PMH hospitals selected for the present study.

The data resulting from the questionnaire application were collected for three months, between 23 June and 23 September of 2021. A total of 2077 of the 3104 responses received were considered valid and distributed as described in Table 1. Invalid responses were due to the fact that they were incomplete, indicating the unwillingness to respond or just curiosity to see what the survey was like.

For the validation of the scales and to test the goodness-of-fit of the final model, 384 questionnaires (test sample) were randomly chosen following a systematic approach with an alternating k between 5 and 6 (Table 1).

### 2.2. Variable Measures and Measurement Model

The literature identifies five models that aim to measure the quality of health services, namely: Donabedian’s, SERVQUAL, HEALTHQUAL, PubHosQual, and HospitalQual [61]. Based on the works of Donabedian [25] and Parasuraman et al. [43], Camilleri and O’Callaghan [62] developed the HEALTHQUAL model, which may be considered an adaptation of the SERVQUAL model, contemplating a set of items for measuring the quality of health services, based on the type of service and the user. In this study, the HEALTHQUAL model adapted and proposed by Lee [63] was used because it was considered an excellent approach to the Portuguese context because of its applicability to hospitals and its suitability to measure the quality of health services based on the patient’s view. This measurement model includes five dimensions: (1) improvement in health care services; (2) tangible quality aspects; (3) efficiency quality aspects; (4) safety quality aspects; (5) empathy quality aspects. Considering the work developed by Lee and Kim [64], it was decided to adapt the HEALTHQUAL scale to measure the quality perceived by the Portuguese users.

Lee’s [63] original scale presented 26 questions with the response format on a 5-point Likert-type scale. Many of the items were not well adapted to this type of response in Portuguese, implying scales between ‘Very bad’ and ‘Very good’ or between ‘Very small’ and ‘Very large’. So, to validate the scale for the Portuguese population, respecting the conceptual approach used by Lee [63], the dimensions were redefined, and the items reframed and converted into statements, which allowed for a response on the classic Likert scale (1 = Strongly disagree; 2 = Disagree; 3 = Neither agree nor disagree; 4 = Agree; 5 = Strongly agree). It is essential to be parsimonious so that the scale would be feasible in obtaining answers and exhaustive in its content validity. Thus, and after the necessary corrections and pre-tests that were performed on 20 users, we reached the final 25 questions to assess the agreement for the five dimensions proposed by the HEALTHQUAL model [63], divided into the following dimensions (Table 2):

The ‘Empathy’ dimension includes items to evaluate the attitude of the hospital’s clinical staff towards better serving patients, actively listening and reflecting on their emotions while providing health care services. The ‘Space and Environment’ dimension refers to the patient’s perception of the quality of the facilities, the physical environment, and mobility. The dimension ‘Effectiveness’ comprises items related to the hospital’s ability to achieve its treatment goals and good service to the population, ensuring the effectiveness of the services provided. The ‘Efficiency’ dimension refers to the rational and parsimonious use of resources, avoiding waste at all levels in providing care services to users. Finally, the dimension ‘Results of the use of hospital services’ refers to the efforts of the clinical staff regarding processes, communications, and interactions with users, and the result perceived by users regarding the improvement of their clinical situation.

The process followed to assess and validate the two scales implied choosing a random sample of 384 valid responses to assure an estimation power of around 85% [65]. Then, we applied the model to all data to verify if it explains what is happening in a larger population sample.

First, the analysis showed that there is no multivariate normality. We chose estimators that do not present this assumption, as is the case, in the AMOS, of the SLS (Scale-free Least Square) or the ADF (Asymptotically Distribution Free).

Second, it was verified whether the data would be adjusted for carrying out factor analyses [66], using the KMO statistic (Kaiser–Meyer–Olkin), the Bartlett sphericity test, the determinant of the R-matrix (of correlations), and the ‘Reproduced’ option that gives a summary of the differences between the correlation matrix based on the model and the correlation matrix based on the actual data. Following Hair et al. [67], a confirmatory factor analysis (CFA) was performed to determine if the dimensions predicted in the original scale were maintained. It used the principal axis factoring method with Varimax rotation, allowing the natural correlation between the dimensions of a latent variable but trying to distinguish them in an orthogonal way.

The values found were as follows: KMO = 0.971 (optimal); Bartlett’s sphericity test: *χ*^2^_(300)_ = 9038.662 with *p* < 0.001 (optimal); correlation matrix determinant = 3.159 × 10^−11^ (very low value, indicating excessive multicollinearity); and the analysis of the calculated residuals between the observed and reproduced correlations showed that there are six (2.0%) with absolute values greater than 0.05 (optimal). In short, we had a problem of excessive multicollinearity that needed to be controlled. As for the number of factors extracted, we used the Kaiser–Guttman criterion, which implies choosing only those with an eigenvalue greater than 1. The variance accounted for by it is greater than the standardized mean variance of all items, therefore equal to 1. It was found that there would only be three factors according to the Kaiser–Guttman criterion, explaining 71.04% of the total variance.

Analysing the distribution of the items, they do not follow the dimensions indicated in the original study by Lee [63]. Then, the scale was purified by eliminating the items loaded in different factors and the items that presented correlations with other variables greater than 0.8. In the end, a new CFA showed that only two factors had eigenvalues greater than 1, explaining 69.14% of the total variance, with all the loadings higher than 0.62 (Table 3).

One can notice that the new dimension, ‘Clinical staff effectiveness’, gathers all the items related to the appreciation of the clinical staff quality that was dispersed by Lee’s [63] dimensions of empathy, effectiveness, efficiency, and results. The other dimension remains the same.

Based on the core competencies that determine the conceptual model of health literacy, six questions were added to the questionnaire that assesses the variable ‘Literacy Level’ (Table 4). Respondents answered using a 5-point Likert scale: 1 = Strongly agree; 2 = Agree, 3 = Neither agree nor disagree; 4 = Disagree; 5 = Strongly disagree.

The values found were as follows: KMO = 0.60 (satisfactory); Bartlett’s sphericity test: *χ*^2^_(15)_ = 553.456 with *p* < 0.001 (optimal); determinant of the correlation matrix = 0.233 (optimal); and the analysis of the calculated residuals between the observed and reproduced correlations tells us that there are two (13%) with absolute values greater than 0.05 (optimal). The three predicted factors would be acceptable, explaining 77.33% of the total variance. The scores obtained by CFA for the literacy level were calculated to study its moderating impact on the final model.

The assessment of the reliability of the scales was performed using several techniques: the corrected item–total correlations (>0.3); the average of the inter-item correlations (>0.5); and Cronbach’s alpha (>0.7) [69]. It was also calculated the composite reliability (CR), the mean of the variance extracted from the latent variables [67,70], as well as the average variance extracted (AVE) that reflects the global amount of variance in the indicators accounted for by each factor (latent). The AVE must be greater than 0.5 [70], and the CR greater than 0.6 [71] or, preferably, 0.7 [67].

The summary in Table 5 shows that all scales are reliable.

However, reliability is a necessary but not sufficient condition for validity [72]. The validity of constructs (latent variables) refers to how a set of items reflects the theoretical construct intended to measure.

In addition to the fact that these scales already exist in the literature (Healthqual—[63]; literacy level—[68]), content validity (representativeness of the items concerning the theoretical domain of the constructs) and face validity (language and format of responses and instructions) were assessed by three experts and 20 users, contributing to their improvement and validity. Convergent, discriminant, and nomological validities were evaluated in CFA and model testing [73]. More specifically, after conducting the CFA, various indices of the model’s goodness of fit were evaluated. If the model’s fit is considered acceptable, then it is possible to confirm the convergent validity and the discriminant validity using the CR and AVE criteria [70,74]. The level of significance and the magnitude of the relationship between the construct of interest and the other constructs of an established theoretical framework will provide evidence of the nomological validity of the measurement [75].

Convergent validity requires that the items (manifest variables) chosen to measure the constructs (latent variables) are associated. That is, they share a high proportion of variance in common. We can make this assessment through the value of the factor weights (Table 3) and the values presented in Table 5 about the reliability analysis, the AVE (>0.5), and the value of the CR (>0.7) [70,76]. MVE > R^2^ demonstrated discriminant validity for each pair of latent variables.

Criterion-related validity has to do with the degree to which the construct is related to a criterion variable, predicting the variations of another related variable. In the present study, since the variables were measured simultaneously, this validity is assessed by concurrent validity. However, it is necessary to assess whether there may be a second-order construct–users’ perceived quality. To this end, the two models were compared with the AMOS: with two first-order factors and with the second-order factor [77], verifying that we have the same value of *χ*^2^ statistic (=265.649) and degrees of freedom (89), so both models represent the same reality, as such, we can use the second-order latent variable referring to the ‘Quality perceived by the users’. This validity was demonstrated and presented in the results obtained with the model in the results and discussion section.

The measurement of the variable ‘Satisfaction’ will allow comparing the users’ perceptions of their level of satisfaction and determining whether there are significant differences according to the type of management of the hospitals that assist them. The satisfaction variable was operationalised by using a single question related to the services provided by the hospitals and which has been used in other studies, e.g., [78,79]. Thus, the statement is ‘Overall, I am very satisfied with this hospital’, which will be answered on a broader scale between 1 = absolutely satisfied to 10—absolutely dissatisfied.

Five items were also used to cover various socio-economic characteristics of the respondent population: gender (U1—male or female), age (U2—18–24; 25–34; 35–44; 44–54; 55–64; more than 65), education (U3—elementary school; secondary school; higher education), and annual gross income, considering the 2020 National IRS (U4—≤EUR 7,112; EUR 7,112 up to EUR 10,732; EUR 10,732 up to EUR 20,322; EUR 20,322 up to EUR 25,075; EUR 25,075 up to EUR 36,967; EUR 36,967 up to EUR 80,882; ≥EUR 80,882).

All the variables were included in a measurement model represented by a path diagram in Figure 1.

This structural equation model envisages that the items are all reflective, constituting the manifest variables of the latent variables or constructs of the model. It is also assessed that the level of literacy, age, education, and income may be moderating variables of the relationship between the quality perceived by the users and their satisfaction with the hospital’s services.

### 2.3. Data Collection Process

The questionnaire design was adjusted so that all questions on the measures of the scales were mandatory, thus solving the problem of non-responses [80]. Considering the limitations imposed by the General Data Protection Regulation, obtaining a list of the users of the sample hospitals was not allowed, which makes the use of a simple random sampling process unfeasible. Thus, the randomness of the sample stems from the free choice of each user of the selected municipalities to answer the questionnaire in digital format and accessible online.

In order to maximise the response rate, the councillors responsible for health in each municipality were contacted to encourage citizens’ responses. Thus, the questionnaire was published on the digital channels of each municipality, namely on the institutional websites, on the official Facebook and LinkedIn channels, and on other channels used in each municipality to contact citizens.

The municipalities of the hospitals under study were absolute admissibility criteria, as they were filtered by the first question of the questionnaire. To validate scales and to obtain a representative sample of users, it was decided that the minimum number of responses should be at least 200 per hospital, which ensures a confidence level above 95% and a test power above 80% [65]. For data entry, editing, review, and analysis, IBM SPSS 26.0 and AMOS 26.0 software (IBM, New York, NY, USA) were used.

### 2.4. Ethical Issues

Concerning data protection information, data were collected, processed, and analysed following the General Data Protection Regulation (EU) 2016/679 of the European Parliament and of the Council of 27 April 2016 on the protection of natural persons about the processing of personal data and on the free movement of such data.

## 3. Results and Discussion

The first research question was answered by calculating the mean level of satisfaction with PPP hospitals, which was 5.25 (SD = 2.31). The users rated it slightly positive, i.e., above the average value of 5.5. This variable has only an excess of kurtosis without skewness, but the sample number of participants allow to perform a *t*-test to answer the second research question, which showed a significant statistical difference (*t* = −10.63; *p* < 0.001) between the average rates of PPP hospitals (5.25) and PMH (6.33; SD = 2.32), also confirmed by the non-parametric Mann–Whitney *U* test (*U* = 398,762; *p* < 0.001). Since the standard deviations and sample size are similar between the groups, we calculated the Cohen’s *d* effect size, which presented the value of 0.47, which can be considered a medium one [81]. Note that the users’ rate of PMH is slightly negative because it is above the average score.

These results do not corroborate the theory proposed by Hsiao [82] and Maynard [83]. They consider that the allocation of management to a private partner of a public hospital negatively affects the satisfaction and assessment of the quality of service provided by the hospital. However, we cannot forget the idea that Padma et al. [84] conveyed, who argued that the private partner is conditioned concerning its effectiveness. This conditioning comes from the possible limitations at the level of the resources contractually made available by the public partner, which may not meet the demands imposed by the population, and end up being more demanding when it comes to handing over the management of the hospital to a private partner. Nevertheless, there is a lower satisfaction among the users of PMH. Thus, our results seem to support the idea conveyed by Tang et al. [29] and Yamout and Jamali [1]. They argued that public services might be more satisfactorily delivered through the appropriate use of the private sector’s skills, experience, technology, and innovation.

It should be noted that the lower user satisfaction regarding the use of PMH can be explained by the significant decrease in public resources allocated to health systems, the migration of health professionals to the private sector and abroad, the increase in the number of patients and the complexity of diseases [85]. These factors have a negative impact on the quality of clinical services and, consequently, on the performance of PMH. Moreover, the EU [86] argued that user satisfaction rates are often high in PPPs, which is supported by our results.

Regarding the third research question, the structural model (Figure 1) presents good goodness of fit indicators (Table 6). Thus, the model represents quite well the data on the test sample and the complete sample of the study, and the values of the weights (regression weights or loadings) between the substantive variables are all statistically significant (*p* < 0.001).

The value of the chi-square statistic is always inflated as the sample size increases and when there is no multivariate normality [76]. The higher the chi-square, the worse the model fit. The chi-square test null hypothesis is that the model fits the data, which in this study is rejected for the reasons mentioned above. It is more important to assess the value of the *χ*^2^/df ratio, which is considered suitable for values below two, and acceptable for values below five [87].

All model variables were measured using a single method—survey—leading to a possible bias called common method variance (CMV). To study this phenomenon, Baumgartner and Weijters [88] point out, after analyzing all the techniques used by the researchers, that the best approach would be to use a confirmatory factor analysis with the introduction of a method marking variable. This technique will be presented in Table 7, following the guidelines of Podsakoff et al. [89], Richardson et al. [90], Simmering et al. [91], and Williams et al. [92]. The marking variable will be the literacy level, as it has nothing to do with the concepts of the latent variables of the quality perceived by users.

There is evidence of CMV between the indicators of the substantive variables and the marker (method-C; *p* < 0.05), with the CMV being the same for all indicators (method-*U*; *p* < 0.01). However, CMV does not bias the relationships between substantive variables (method-R; *p* = 0.956). Although there is some effect of the CMV, it is not enough to interfere with the results on the relationships between the substantive variables of the model.

In Table 8, the results obtained with the final estimation of the model are presented, where the positive impacts that exist between the variables under study can be verified, as expected. These results corroborate what has been concluded by other studies carried out in other countries, e.g., [62,63,64].

Given the results obtained, it is possible to conclude that the model is valid, thus enabling us to answer the fourth research question: does literacy level, age, education, and income moderator variables of the relationship between the quality perceived by the users and their satisfaction?

The model was tested according to the possibility that the literacy level (LL) could be a moderating variable. LL was dichotomized according to the average of the scores obtained by the regression method in the context of factor analysis (average = 0). It presents 1015 people (48.9%) who have a LL below the average and 1062 (51.1%) above average.

The LL is a moderating variable in the relationship between perceived quality and users’ satisfaction (Table 9). A unitary variation of the standard deviation in the quality perceived by users with lower literacy levels implies a more significant variation in the standard deviation of users’ satisfaction.

These results may have several interpretations. Users with a higher degree of literacy tend to consider many more variables to assess their degree of satisfaction with the hospital they used. These include knowing or not which management model is applied in hospitals. Therefore, what is evaluated by the quality scale has less weight in the degree of satisfaction. Siemiatycki [93] argued that it is essential to develop processes for engaging the population, namely the dissemination of data and performance of the PPP hospitals, to avoid formulating negative opinions regarding the application of the model. Any additional effort to better inform the population with a lower level of literacy could significantly impact their perception of quality and, consequently, their satisfaction. Similar results appeared in other studies, such as Canada’s lack of ‘external’ transparency. A study by Dalton-Jez et al. [4] concluded a communication gap with the population when a new hospital was opened under a PPP regime. Skietrys et al. [34] also identified communication as one of the fundamental problems, which may give rise to assumptions regarding the benefits of the PPP model among the population. Thus, we may conclude that the Portuguese population is not adequately informed and enlightened regarding the PPP model in the health area.

The model was tested on the possibility that age could be a moderating variable. The comparison between individuals older than 45 years with those younger than this upper limit of the median bracket shows no statistically significant differences between the loadings in the two groups (Table 10). These results are in line with research carried out by Mummalaneni and Gopalakrishna [94], which states that hospitals should not be concerned with the sociodemographic characteristics of users when trying to improve their satisfaction with health care. Hospitals should focus their attention on attributes such as the quality of care. Thus, these authors conclude that the influence of age on user satisfaction is negligible, as our results suggest. Miljanović [95] also concluded that there are no significant mediating effects of sociodemographic variables (age) on the correlation between the hospital used and satisfaction with health care.

The model was also tested on the possibility that Income could be a moderating variable. In comparing individuals with incomes above EUR 1,452 (upper limit of the median bracket) with those with a lower income, there are no statistically significant differences between the loadings in the two groups (Table 10). However, this variable can be considered a mediator for a *p*-value < 0.1. This result happens with all hospitals. However, when evaluating PPP and PMH (Table 10), only differences in PPP could be considered significant and with *p* < 0.1.

Given that PPP hospitals provide clinical services at no cost to users (a service that tends to be free), as do PMH, it seems natural that income does not have a moderating role in terms of users’ perception of quality and satisfaction. Our results contrast with the research carried out by Mummalaneni and Gopalakrishna [94], where they found that income seems to have a moderating impact on user satisfaction with the hospital used. However, their study was carried out in a different context, both in space and time.

Finally, the model was tested as a function of the level of education (basic, secondary, and high) to verify whether this variable would be a moderating variable in the relationship between perceived quality and user satisfaction (Table 10). In comparing individuals with different levels of education, there are no statistically significant differences between the loadings in the three groups (Table 10). However, this variable can be considered a mediator for a *p*-value < 0.1. In a more detailed analysis, it is found that only users who evaluated PPP hospitals showed statistically significant differences (*p* < 0.001) between the three groups of education. As with the level of literacy, the lower the level of education, the more significant the impact between perceived quality and users’ satisfaction with PPP hospitals. The more educated people are, the less impact there is between the perceived quality of clinical services at PPP hospitals and the level of satisfaction with these services. In practice, as we had already seen about the literacy level, people with more education have a higher level of demand, which is more evident in PPP hospitals. According to Ng et al. [55], citizens are more cautious about the quality of services when hospitals or health services are provided through the PPP model, which may explain the differences verified in the evaluation of PPP hospitals related to the level of people’s schooling. However, the literature shows that education adds little explanatory value in terms of users’ satisfaction with health services, e.g., [96,97]. These studies support our global results about the moderation role of the level of education in the relationship between perceived quality and users’ satisfaction.

Like other studies, this one presents some limitations, such as the fact that data was collected online during the pandemic crisis. Although the scientific community widely accepts the questionnaire technique, it is not perfect for gathering information from all kinds of people; it is necessary to explore other approaches to hospital users, namely qualitative means. Future work could investigate these issues from the point of view of health professionals from PPP hospitals and PMH. It could also be interesting to study the behaviours of patients and doctors and the doctor-patient relationships in both types of hospitals.

## 4. Conclusions

The literature presents a significant gap in knowledge due to the scarcity of research analysing the impact of the PPP model regarding service quality and satisfaction, and there is limited guidance for decision-makers on the circumstances in which this model is likely to produce good results. The literature also suggests a greater focus on rigorous evaluations of the PPP model, emphasising quality as a performance criterion. This study may mitigate this gap by presenting data on perceived quality and users’ satisfaction with using hospital units that operate or have operated under this model.

Concerning users’ satisfaction, it was concluded that PPP hospitals satisfy the population (first research question), with a significant statistical difference in relation to the lower level of satisfaction with PMH (second research question). The scale Healthqual was assessed for the Portuguese population, resulting in its adaptation and validation. This scale measures the perceived quality of hospital services, namely the clinical staff effectiveness and the space and environment provided by those institutions. Another scale was validated that measures the population’s literacy level. Then, a structural equation model was proposed to evaluate the relationships between perceived quality and users’ satisfaction, which showed an excellent adherence to data collected on the population served by the four PPP hospitals and the four PMH (third research question). Thus, the results show a significant impact of perceived quality on users’ satisfaction with the hospitals. Moreover, a moderating effect of literacy level on this impact, being people with less literacy can benefit more from information about hospital management (fourth research question). A similar result appeared with people with less education, but in this case, only in PPP hospitals. Given that most of the population referred to this lack of government information about the hospital management models, this type of moderation between perceived quality and users’ satisfaction is expected. It is also concluded that more educated users are more demanding regarding PPP hospitals and PMH.

Illiteracy regarding the PPP model applied to the health sector can lead to feelings of injustice, influenced by the information transmitted by electoral programmes or communication campaigns of the various political forces. In other words, it is clear that for the Portuguese population, which is poorly literate on the subject of PPPs, all information that portrays negative results of the model in health tends to generate, according to the theory of social exchange, a feeling of distributive injustice. Moreover, the current Portuguese Government’s decision not to renew PPP contracts in the health sector might be questionable since some performance indicators [60] and users’ satisfaction with the services point to it being a beneficial management model for the state and taxpayers.

In what concerns the other two possible moderating variables, age and income, the results showed that they do not seem to influence the relationship between the substantive variables of the model (fourth research question).

Our study is the first to present comparative results on users’ perceptions of PPP and PMH hospitals. We hope policymakers can also consider these results when evaluating future PPP contracts in the health sector.

## Figures and Tables

**Figure 1 ijerph-19-08188-f001:**
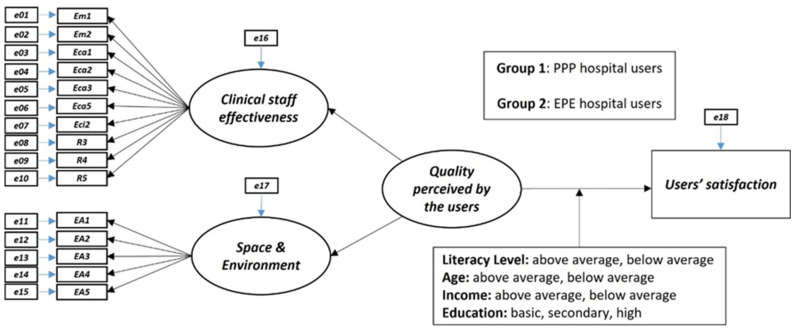
Path diagram of the model.

**Table 1 ijerph-19-08188-t001:** Distribution of valid answers.

Hospitals	Frequency	Percentage	Test Sample *
Braga PPP	233	10.1	43
Cascais PPP	222	9.6	41
Loures PPP	304	13.2	56
Vila Franca de Xira PPP	274	11.9	51
** *Sub-total* **	** *1033* **	** *44.8* **	** *191* **
Figueira da Foz PMH	212	9.2	39
Setúbal PMH	210	9.1	39
Leiria PMH	372	16.1	69
Évora PMH	250	10.9	46
** *Sub-total* **	** *1044* **	** *55.2* **	** *193* **
**Total**	**2077**	**100**	**384**

* Frequencies obtained by the application of systematic random sampling.

**Table 2 ijerph-19-08188-t002:** Items of the variable perceived quality.

Quality Perceived by the User	ID	Questionnaire Items
**Empathy**	Em1	The clinical staff is polite and friendly.
	Em2	The clinical staff listens and understands the users.
	Em3	The clinical staff easily understands my health situation.
	Em4	The clinical staff quickly understands my needs.
	Em5	The clinical staff knows how to put themselves in my place and understand my problems.
**Space and environment**	EA1	The facilities of this hospital seem to me to be adequate to the needs of the county’s population.
	EA2	The hospital’s physical facilities are visually appealing.
	EA3	The overall cleanliness level of the hospital is adequate.
	EA4	The Hospital provides a comfortable and safe environment for users.
	EA5	Mobility within the hospital is relatively easy.
**Effectiveness**	Eca1	The clinical staff was rigorous in hygiene care and protection.
	Eca2	The doctors made an accurate diagnosis of my pathologies.
	Eca3	The nurses did not make mistakes in their work.
	Eca4	I felt confident about the professional performance of the entire clinical staff.
	Eca5	I have great confidence in the ability of the doctors in this hospital.
**Efficiency**	Eci1	I feel that the clinical staff avoids unnecessary use of medications.
	Eci2	I feel that the clinical staff strives to apply only those methods of treatment that are strictly necessary.
	Eci3	I feel that the hospital’s work processes are clear and concise.
	Eci4	I feel that the number of employees is adequate for the needs of the users.
	Eci5	The hospital efficiently manages the time I spend in the hospital (e.g., conducts appointments/exams on time).
**Results of the use of hospital services**	R1	I feel that the health care was appropriate for my medical situation.
	R2	My health has improved after the treatment in this hospital.
	R3	I feel that I have improved a lot because of being treated in this hospital.
	R4	The clinical staff gave me all the explanations to prevent other related diseases.
	R5	The degree of effort and willingness of the clinical staff to prevent disease is high.

**Table 3 ijerph-19-08188-t003:** Dimensions and item loadings of the variable perceived quality.

**Clinical Staff Effectiveness**	**Em1**	**Em2**	**Eca1**	**Eca2**	**Eca3**	**Eca5**	**Eci2**	**R3**	**R4**	**R5**
	0.759	0.795	0.626	0.807	0.621	0.867	0.684	0.821	0.771	0.724
**Space and environment**	**EA1**	**EA2**	**EA3**	**EA4**	**EA5**					
	0.771	0.814	0.643	0.708	0.645					

**Table 4 ijerph-19-08188-t004:** Items of the variable literacy level.

Literacy Level	ID	Questionnaire Items
**Access competence**	GLA1	I have already looked for information about the hospital management model applied in the health care unit in my area of residence.
	GLA2	I consider that the information made available by the government regarding hospital management models is enough.
**Understanding competence**	GLC1	I know well what a public–private partnership model in hospital management is.
	GLC2	I know well the differences between a public–private partnership management model and an exclusively public management model.
**Evaluate and apply information competence**	GLI1	The management model applied influences my choice, concerning the health unit (Hospital).
	GLI2	I have a positive opinion about the involvement of the private sector in the management of Public Hospitals.

Based on [68].

**Table 5 ijerph-19-08188-t005:** Reliability and validity analysis.

Scales	Cronbach α	Minimum of the Corrected Item–Total Correlations	Average of the Inter-Item Correlations	Compositive Reliability	Average Variance Extracted	R^2^
Clinical staff effectiveness (CSE)	0.948	0.668	0.647	0.966	0.738	CSE-SE (0.676) LL-SE (0.095) LL-CSE (0.107)
Space and Environment (SE)	0.880	0.661	0.594	0.936	0.746	
**Service quality**	**0.946**	**0.548**	**0.539**	**0.976**	**0.733**	
Literacy Level (LL)	0.757	0.387	0.509	0.834	0.643	

**Table 6 ijerph-19-08188-t006:** Indicators of the goodness of fit of the final model with test sample and complete sample.

Indicators	N = 384	N = 2077	Criteria	Indicators	N = 384	N = 2077	Criteria
** *CMIN* **	162.391	365.963	Smaller	** *NFI* **	0.796	0.897	>0.9
** *df* **	90	90	-----	** *RFI* **	0.728	0.862	>0.9
** *p-value* **	0.000	0.000	>0.05	** *TLI* **	0.857	0.892	>0.9
** *CMIN/df* **	1.804	4.066	<2 (5)	** *CFI* **	0.893	0.919	>0.9
** *RMR* **	0.111	0.067	Smaller	** *PNFI* **	0.597	0.672	>0.6 (0.8)
** *SRMR* **	0.096	0.052	Smaller	** *PCFI* **	0.670	0.689	>0.6 (0.8)
** *GFI* **	0.914	0.960	>0.9	** *FMIN* **	0.424	0.159	Smaller
** *AGFI* **	0.871	0.940	>0.9	** *RMSEA* **	0.046	0.036	<0.05
** *PGFI* **	0.605	0.635	>0.6 (0.8)	** *PCLOSE* **	0.718	1.000	>0.05

CMIN (chi-square statistic); df (degrees of freedom); RMR (root mean square residual); SRMR (standardized root mean square residual); GFI (goodness of fit index); AGFI (adjusted goodness of fit index); PGFI (parsimonious goodness of fit index); NFI (normed fit index); RFI (relative fix index); TLI (Tucker–Lewis Index or NNFI—non-normed fit index); CFI (comparative fit index); PNFI (parsimonious normed fit index); PCFI (parsimonious comparative fit index); RMSEA (root mean square error of approximation); PCLOSE (H_0_: RMSEA < 0.05).

**Table 7 ijerph-19-08188-t007:** Common method variance study.

Modelo	*χ* ^2^	df	*χ*^2^/df	RMSEA (*p*-Value)	SRMR	CFI	RFI	GFI	Δdf	Δ*χ*^2^	LR of Δ*χ*^2^	Comparison Model
**CFA with Marker**	468.696	132	3.551	0.082	0.057	0.932	0.893	0.873				
**Baseline preparation**	468.696	138	3.396	0.079	0.057	0.933	0.898	0.873	6	0.000	1.000	
**Baseline**	482.700	140	3.448	0.080	0.074	0.931	0.896	0.870	2	14.004	**0.001**	
**Method-C**	477.793	139	3.437	0.080	0.061	0.931	0.897	0.871	1	4.907	**0.027**	Baseline
**Method-U**	444.684	125	3.557	0.082	0.054	0.935	0.893	0.880	14	33.109	**0.003**	Method-C
**Method-R**	444.687	126	3.529	0.081	0.054	0.935	0.894	0.880	1	0.003	**0.956**	Method U

LR—likelihood ratio.

**Table 8 ijerph-19-08188-t008:** Direct effects on the final model.

Relationships	Loadings
Perceived quality → Space and Environment	0.842
Perceived quality → Clinical staff effectiveness	0.924
Perceived quality → Users’ satisfaction	0.946

**Table 9 ijerph-19-08188-t009:** Study of the moderator effect of literacy level.

Models	Chi-Square	Degrees of Freedom	*p*-Value	Invariant?
Not restricted	502.993	180		
Restricted	533.928	195		
**Difference**	**30.935**	**15**	**0.009**	**No**
		**All hospitals**	**PPP hospitals**	**PMH**
**Loadings of Perceived quality** **on Users’ satisfaction**	** *Below average* **	0.968	0.951	0.982
	** *Above average* **	0.943	0.921	0.938

**Table 10 ijerph-19-08188-t010:** Study of the moderator effect of age, income, and education.

Models (Age)	Chi-Square	Df	*p*-Value	Invariant?
Not restricted	478.819	180		
Restricted	480.056	181		
**Difference**	**1.237**	**1**	**0.266**	**Yes**
**Models (Income)**	**Chi-Square**	**Df**	***p*-value**	**Invariant?**
Not restricted	488.747	180		
Restricted	491.686	181		
**Difference**	**2.939**	**1**	**0.086**	**Yes**
**Models (Education)**	**Chi-Square**	**Df**	***p*-value**	**Invariant?**
Not restricted	655.323	270		
Restricted	660.480	272		
**Difference**	**5.157**	**2**	**0.076**	**Yes**
**Loadings of perceived quality** **on users’ satisfaction**	**Age**	**All hospitals**	**PPP hospitals**	**PMH**
	*Below 45 years*	0.957	0.933	0.971
	*Above 45 years*	0.948	0.946	0.937
**Loadings of perceived quality** **on users’ satisfaction**	**Income**	**All hospitals**	**PPP hospitals**	**PMH**
	*Below 1452€*	0.955	0.948	0.957
	*Above 1452€*	0.959	0.923	0.967
**Loadings of perceived quality** **on users’ satisfaction**	**Education**	**All hospitals**	**PPP hospitals**	**PMH**
	*Basic*	0.964	0.963	0.937
	*Secondary*	0.942	0.944	0.930
	*High*	0.971	0.921	0.971

## Data Availability

Database is available at: Carvalho, João (2022), ‘Survey on Health Quality Service’, Mendeley Data, V1, doi:10.17632/k2556c5cr6.1.

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
