# Peer review of "Perceived Quality and Users’ Satisfaction with Public–Private Partnerships in Health Sector"

_ijerph, 2022, doi:10.3390/ijerph19138188_

Round 1
Reviewer 1 Report
Please see attached

Reviewer 2 Report
Review of “Perceived Quality and Users’ Satisfaction with Public-Private Partnerships in Health Sector”
This study aims to empirically assess the satisfaction of the Portuguese population regarding the Public-Private Partnerships model implemented in four Portuguese hospitals; to evaluate the quality of service provided by hospitals operating on the PPP model; and to analyse the possible moderator effect of several variables (literacy level, age, education, and income) of the population (Portuguese) regarding the PPP model in health.
The research appears sound (based on 2,077 questionnaires), well-structured, and correctly described in the paper. Also the past literature was correctly considered. I think these kinds of research are important for scholars and, most of all, for policy makers. Although I think the paper is already good enough for the publication, I have few suggestions to provide to the authors:
- I suggest to the authors to proofread the paper. For example, in the abstract the last part of the following sentence is unclear/incorrect to me “This study aimed to empirically analyse the degree of satisfaction of the Portuguese population regarding the service quality provided by PPP and Public Management hospitals (PMH) within a structural equation model, and verify the moderating effects of people's literacy level, age, education, and income, moderate their opinions.”.
- Maybe a figure in the results section showing directly and simply the significant relationships discovered may help the readers.
- As future development of the work, I would propose in the conclusion section to study also the behaviors of patient/customers and doctors in the hospital and the doctor-patient relationships. In this direction, I can suggest the following citations: - Stefanini, A., Aloini, D., Gloor, P., Pochiero, F. (2021). Patient Satisfaction in Emergency Department: Unveiling Complex Interactions by Wearable Sensors. Journal of Business Research, 129, 600-611. DOI: 10.1016/j.jbusres.2019.12.038. - Kawamoto, E., Ito-Masui, A., Esumi, R., Imai, H., & Shimaoka, M. (2020). How ICU patient severity affects communicative interactions between healthcare professionals: A study utilizing wearable sociometric badges. Frontiers in Medicine, 7, 606987.
Reviewer 3 Report
This is a very interesting and well performed paper. However, prior to publishing, it requires several adjustments.
My detailed comments are as follows:
1. In abstract (lines 22-23) the authors stated: "one wonders 22 why the Government insists on avoiding the contracts' renewal" - the paper is focused mostly on patients satisfaction, while many aspects important for hospitals are not being analyzed. As a result the authors have no scientific right to claim such statement. Please delete it or if not, support scientifically such thesis in paper.
2. Line 70. Reference 15 is valid only for Russia. Please provide the comment that the statement in lines 69-70 is valid for Russia or provide another reference, which gives the right to state hat it is a globally acceptable statement.
3. Lines 95-97. The authors provide "patient perceptions of health care have been predominantly accepted as the critical indicator for measuring health care quality and as a critical component of improving clinical performance and effectiveness". The reference 24 is a paper focused on patients satisfaction and its conclusion is not as cited above. Please justify better why do you claim that patient perceptions of health care is critical component of improving clinical performance and effectiveness or restate that sentence.
4. Line 219. Please comment more why do you consider 4 PPP hospitals can be compared with selected 4 PMH.
5. I recommend to consider the split of discussion from results. If not in this paper, than in the future ones. It is more transparent when readers can see pure results obtained in the paper and discussion separate.
6. Lines 604-609 should not be presented as conclusion. This should be provided in discussion section.
7. The study developed 4 research questions. I would expect the authors to mention them and answear directly. (I confirm indirectly the questions have been well adressed.
